# Experimental and Numerical Study of Shear Interface Response of Hybrid Thin CFRP–Concrete Slabs

**DOI:** 10.3390/ma14185184

**Published:** 2021-09-09

**Authors:** Amir Mahboob, Lluís Gil, Ernest Bernat-Maso, Amir Reza Eskenati

**Affiliations:** 1Strength of Materials and Structural Engineering Department, Technical University of Catalonia, ESEIAAT, C/Colom 11, TR45, 08222 Terrassa, Spain; lluis.gil@upc.edu (L.G.); ernest.bernat@upc.edu (E.B.-M.); amir.reza.eskenati@upc.edu (A.R.E.); 2Serra Húnter Fellow, 08222 Terrassa, Spain

**Keywords:** CFRP, shear connection, hybrid slabs, experimental, numerical

## Abstract

Hybrid slabs made of carbon-fiber-reinforced polymer (CFRP) and concrete provide a solution that takes advantage of the strength properties of both materials. The performance of the system strongly depends on the CFRP–concrete interaction. This study investigates the shear behavior in the interface of the two materials. Eight full-scale experiments were carried out to characterize the interface shear response of these hybrid elements using different connection solutions. An untreated surface is compared to a surface with aggregates, with a novel system comprising a flexible, straight glass fiber mesh and an inclined glass fiber mesh. The experimental results show that the fabric connection improves the friction between materials and is responsible for the pseudo-plastic performance of the specimens. The inclined mesh produces a more uniform tightening effect compared to the straight mesh. In simulations via the finite element method, we used an adjusted frictional model to reproduce the experiments.

## 1. Introduction

Within the last six decades, many reinforced concrete structural elements have been developed under different design code regulations and using different construction techniques. They require maintenance, retrofitting, or even strengthening; see, for example, Colajanni et al. [1]. In addition, some factors such as earthquakes cause damage to these structures, so repair interventions are also required. For this purpose, several experiments on FRP retrofitting/strengthening systems have been carried out in recent years; see reviews in Siddika et al. [2] and Naser et al. [3]. This is because of the benefits of these composite materials, such as their light weight and excellent tensile performance. Hence, FRP performs well at strengthening reinforced concrete structures subjected to a wide range of loads, including seismic ones, as detailed in Kunwar et al. [4], and impact, as detailed in Pham and Hao [5].

Regarding hybrid beams, they are mainly used for bridge applications; see Zou et al. [6]. Hulatt et al. [7,8] studied two kinds of FRP–concrete hybrid beams and investigated their failure mechanism parameters. Both short-term and long-term loadings were applied to the hybrid beams. Brittle failure was observed in most cases. In other studies, the behavior of a highway overpass composite bridge was studied by Mieres et al. [9] and Gutiérrez et al. [10]. The girders of the bridge were reinforced with carbon fiber with a low elastic modulus and high strength. After quasi-static tests, some failure modes occurred, including joint separation and support diaphragm buckling. Furthermore, Hall and Mottram [11] investigated the bending behavior of FRP–concrete beams reinforced with two or four T-shaped FRP profiles. Shear failure was observed at the top of the unreinforced concrete.

Given that most of the FRP–concrete hybrid structures use FRP pultruded beams, it is important to highlight the work by Bank [12], who investigated the experimental characterization of FRP pultruded beams and presented a methodology for evaluating the Young’s modulus and shear modulus of each specimen by using the Timoshenko beam and three-point bending test. Barros and Lebre [13] and Roberts and Al-Ubaidi [14] focused on the stiffness parameters obtained from bending and tensile tests. The experimental results were confirmed using theoretical equations. Canning et al. [15] tested six concrete hybrid beams. The highest flexural stiffness was found for a specimen reinforced with a bonded FRP plate, whereas the beam reinforced with a bolted connection had the lowest strength, indicating that the mechanical connection of FRPs was less effective than the adhesive connection. Nordin and Taljsten [16] also evaluated the shear strength of two connection methods of reinforcement: bolt connection and epoxy resin adhesive. The results indicated that the flexural stiffness of the epoxy-resin-bonded beam was higher than that of the beam reinforced with the bolt connection. Wang et al. [17] presented a methodology to calculate the ultimate capacity of beams reinforced with composites. They also investigated the failure modes of the specimens. Di et al. [18] focused on the failure mode and the shear capacity of GFRP–concrete composite beams. Their experimental tests showed that the minimum ultimate shear capacity was obtained for the single-bolt connection. Tureyen and Frosch [19] studied the shear strength of concrete beams reinforced with FRP bars. Guadagnini et al. [20] tested three concrete beams that were reinforced with conventional steel and another three specimens reinforced with glass fiber bars. Glass bar reinforcement led to shear failure. The authors recommended that ACI 440 and the Institution of Structural Engineers (U.K.) be taken into account in the design and analysis of FRP concrete elements. Alam and Hussein [21] carried out experimental tests on FRP-reinforced concrete elements without web reinforcement. Their experimental results illustrated that different parameters, including the span-to-depth ratio and the depth of the beam, had a considerable effect on the shear strength of the beams. Dias et al. [22] performed a series of experiments to evaluate the performance of a hybrid solution employing CFRP for the flexural and shear strengthening of the beams. Using the experimental results, the predictive performance of the ACI formulations suggested for the contribution of CFRP systems to the flexural and shear resistance of RC beams was evaluated. Hadhood et al. [23] used the combination of two techniques, externally bonded (EB) and mechanically fastened (MF) CFRP strips with great potential in shear strengthening, aiming to strengthen beams with short CFRP strips using an EB/MF hybrid technique. The use of short CFRP strips resulted in a shear load increase of up to 65 percent over that of the un-strengthened specimen. Also, Siwowski and Rajchel [24] studied an FRP–lightweight concrete hybrid girder for bridge superstructures, and Rajchel et al. [25] developed a novel shear connection system for FRP–concrete hybrid bridge girders. Zou et al. [26] used FRP to study the solution of a truss combined with ultra-high-performance concrete (UHPC) for a hybrid bridge.

Fam and Skutezky [27] investigated the mechanical responses of concrete slabs reinforced with GFRP profiles. This study illustrated the failure of hybrid beams in the connection area, due to low resistance. Moreover, the web buckling phenomenon was observed. Elmahdy et al. [28] and El-Hacha and Chen [29] analyzed some beams constructed from ultra-high-performance concrete (UHPC) reinforced with GFRP shear studs, GFRP boxes, and CFRP sheets or steel-fiber-reinforced polymer (SFRP) sheets. The results indicated that the SFRP sheets led to better performance than the CFRP sheets. Five different methods and experimental databases covering the latest research on one-way hybrid slabs were evaluated by El-Sayed and Soudki [30]. Tavakkolizadeh and Saadatmanesh [31] investigated the behavior of concrete slabs reinforced with CFRP sheets and epoxy adhesive. Their analytical and experimental results showed that epoxy-bonded CFRP caused an increase in the ultimate load capacity and stiffness. A solution of FRP and UHPC concrete was presented in Zhang et al. [32] and [33].

Al-Amery and Roberts [34] introduced a finite element method for nonlinear analysis of the partial shear connection in composite beams. Thevendran et al. [35] used ABAQUS software to simulate composite beams in order to investigate their load-bearing capacity. Sebastian and McConnel [36] modeled some composite beams by using nonlinear finite elements. Yang et al. [37] reported a numerical simulation of FRP-strengthened RC slabs with FRP anchors using the finite element method. Their numerical results indicated that some failures occurred in the study, including critical diagonal crack debonding and plate-end debonding. Kong et al. [38] evaluated the behavior of concrete slabs reinforced with Aramid-fiber-reinforced polymer (AFRP) under a blast load using Ansys software. The numerical response of the RC slabs was affected by certain parameters (AFRP layer, trinitrotoluene (TNT), etc.). Ban and Bradford [39] investigated the flexural behavior of a hybrid beam reinforced with a steel profile and constructed a three-dimensional model using the finite element method. Zheng et al. [40] compared the experimental results and numerical outputs of concrete bridge deck structures. The arch-shaped concrete slabs were reinforced with FRP. Mahboob et al. [41] used a mesh grid as a reinforcement instead of steel bars. This approach significantly increased the load-bearing capacity and ductility of the CFRP–concrete slabs. Gong et al. [42] studied the interfacial stress in FRP–concrete hybrid beams using the finite difference method (FDM) and push-out tests. Moreover, Gong et al. [43] performed an experimental investigation of the natural bonding between FRP and concrete.

Hence, there is much research interest on hybrid FRP–concrete structures, but previous studies mainly addressed bridges or cross sections composed of pultruded beams and concrete. There are few studies specifically conducted on hybrid slabs of thin CFRP laminates and even fewer on the mechanical connection between the materials.

In this paper, we aim to investigate the interface shear mechanical response of hybrid CFRP sheet–concrete slabs characterized by the use of connectors such as bonded particles or a high-strength glass fiber mesh. Experimental and numerical analyses of these elements are included and compared in order to validate and analyze the performance. The connection of hybrid CFRP–concrete using a flexible material is a novel idea that is compared to the traditional roughening method using aggregates. Moreover, it has the advantage of connecting materials without the use of any steel bolts, avoiding the problem of corrosion. Finally, the manufacture of these slabs seems to be very simple and competitive in the construction market.

## 2. Experimental Program

### 2.1. Materials

#### 2.1.1. Concrete

All specimens were cast using commercial (FIASA HS-25) dry concrete (Fiasa, Barcelona, Spain) distributed in 25 kg bags. It was composed of 300 kg/m^3^ of Portland cement with continuous 0 to 12 mm limestone aggregates and plasticizer. It was mixed with a water/cement ratio of 0.6, reaching a compressive strength below 25 MPa.

The strength of the concrete was estimated using a non-destructive Schmidt impact hammer (Screening Eagle, Zurich, Switzerland). Six repetitions were conducted on each specimen to estimate the average value of the compressive strength according to the ASTM method [44]. Table 1 shows the average compressive strength of each specimen.

#### 2.1.2. CFRP Sheets

CFRP sheets were manufactured using the traditional hand layout process on a mold of a thin steel sheet with a top-hat shape. The material was unidirectional carbon fiber Masterbrace FIB 300/50 (MBCC Group, Barcelona, Spain). Each CFRP sheet had three plies with orientation [0°/90°/0°]. Epoxy resin Resoltech 1204 (Resoltech, Marseille, France) was brushed over every ply using a fiber/resin ratio (by weight) of 1. After impregnating the fibers, a counter mold was placed over them, and some weight was applied to it for a 48 h period to guarantee uniform impregnation. Tensile tests were conducted on eight samples extracted from the manufactured CFRP sheets according to the ASTM test [45] with a strain gage attached in order to measure the Young’s modulus. The modulus of elasticity was 45.55 GPa and the ultimate tensile strength was 1120 MPa, calculated from the ultimate load divided by the area of the coupon.

#### 2.1.3. Mesh

For some connections between the CFRP sheet and concrete, we used a flexible mesh made of MapeGrid (G220) glass fiber (MAPEI, Mount Wellington, New Zealand) with a weight of 225 g/m^2^ and an average tensile strength of 45 kN/m. The mesh had a grid distribution of 25 × 25 mm, leaving empty spaces to mechanically lock the poured concrete. The grid was bonded to the CFRP sheet on the flat upper surfaces using a carbon fiber strip and the same resin (Resoltech 1024). The mesh was placed aligning the grid with the nerves of the slab in some specimens (straight specimens), while in others, the relative orientation between the grid and the nerves was 45° (inclined specimens).

#### 2.1.4. Sand and Gravel

Solid particles of different sizes were used to increase the friction interface between the CFRP sheet and concrete. The dimension of those aggregates depended on the size of the specimen to ensure reliable manufacturing and presented a diameter between 0 and 4 mm for sand and between 5 and 12 mm for gravels. The amount of aggregates was measured by weight using 300 g of gravel and 150 g of sand, and aggregates were randomly distributed on the surface of the CFRP sheet. The aggregates were bonded by brushing the surface of CFRP with Resoltech 1024 and dropping them on.

### 2.2. Specimens

Eight hybrid specimens designed for direct shear testing were manufactured. The specimen geometry consisted of two parts with different lengths, as shown in Figure 1. Concrete and CFRP were connected using different solutions. Specimens were labeled according to the connection type and number: N-# when there was no connection system, A-# when there were only bonded aggregates to increase the roughness, A-SM-# when there were bonded aggregates and a straight glass fiber mesh, and A-IM-# when there were bonded aggregates and an inclined glass fiber mesh at 45 degrees from the nerves. Table 1 presents the complete specimen labeling system.

Figure 2 shows the casting mold composed of wood, with two holes for the blocks of concrete over the CFRP. Moreover, the straight mesh is shown in the form of a black grid aligned with the nerves, and aggregates can be seen as small, grey-yellow materials on the bottom area of the CFRP sheet.

### 2.3. Test Setup

A hydraulic jack with manual control (Larzep, Mallabia, Spain) was placed between the two blocks of the specimen. One of the concrete blocks acted as a reaction wall while the other failed under a pushing force. The stability of the system relied on the shear stresses between the two materials. The shear test setup is shown in Figure 3 and Figure 4.

Force was uniformly applied through a steel loading tool (U80 steel profile) (Celsa Group, Barcelona, Spain) that increased the contact area between the jack and the tested block in order to avoid local concrete damage. The applied force was increased gradually, and it was continuously measured with a 10 kN UC9 load cell (Celsa Group, Barcelona, Spain). Two LVDT sensors with a range of 20 mm and 0.02% linearity were placed on each side to measure the relative displacement between the smaller block and the CFRP sheet. A data acquisition system recorded all data simultaneously.

## 3. Experimental Results and Discussion

### 3.1. The Failure Process

All specimens failed at the surface contact of the CFRP and concrete. The smallest block resisted up to a maximum value of force and then physically separated from the CFRP sheet. In the case of samples N and A, the failure was sudden and the load dropped instantaneously. Minor friction due to the roughness of the surfaces remained, but only for very low load values. For samples A-SM and A-IM, the failure was progressive. The load reached a maximum and the connection started to separate, but friction related to the existence of the glass fiber mesh maintained the load value. The ultimate failure occurred with a mixed mode, combining breakage of the fibers and debonding of the mesh (see Figure 5).

The values of failure were pretty different from those of other types of connection based on bolts. Zou et al. [26] observed sudden and fragile failure that happend because of bolt failure or because of FRP flange shear-out. Di et al. [18] and Rajchel et al. [25] observed large plastic bolt deformations in the load–slip curves. The shape of those load–slip curves has some similarity to that of the A-IM samples. It is also interesting to note that Gong et al. [43] observed progressive failure before the bearing stage, due to the loss of the natural bond when using stay-in-place forms (SIP), and they achieved curves similar to those of samples AS-M. Later, when bolts bore loads, the performance was totally different from that of our samples.

### 3.2. Comparison of Results

The force–displacement plots are shown in Figure 6. Force was measured directly and displacement was calculated as the average of both LVDTs. These plots also include the values of the average shear stress and tangential strain. The calculation of the shear stress (in kilopascals) was performed by dividing the value of force by the surface area of the small block, which was 80,000 mm^2^. Moreover, the calculation of tangential strain was performed simply by dividing the average displacement measured with LVDTs (in millimeters) by the length of the small block, which was 200 mm.

From the plots in Figure 6, it is possible to see the differences in the performance before nonlinear behavior was achieved. The interface with no treatment (N case) produced very sparse results. The value of stresses ranged from 53.75 kPa to 0.37 kPa (143 times) for a mean level of strain of 0.004. The reason for this high variability relies on the unpredictable bond capacity of both materials because the samples were manufactured by hand. Basically, there is a frictional mechanism between the concrete and CFRP that depends mostly on the roughness, as Gong et al. [43] noticed in push-out tests. For the A case, differences were smaller and the span of stresses was from 20.87 kPa to 10.63 kPa (2 times) with a mean strain of 0.002. Moreover, the inclined mesh (A-IM case) presented values of stress from 51.25 kPa to 18.5 kPa (2.8 times) with a mean strain of 0.0013. Finally, the stress for the straight mesh (A-SM case) spanned from 60 kPa to 29.86 kPa (2 times) with a strain of 0.004.

From analysis of the stress variation, it is clear that the untreated surface showed an irregular response; therefore, this technique is not recommended for practitioners. It may produce non-conformities in the execution that are more severe than those that occur with hand manufacturing. The surface treatment improved the homogeneity of the results. The mean average stresses presented two groups: Case A (aggregate connections) with a level of 15.75 kPa and Cases A-SM and A-IM (mesh connections) with levels of 44.93 kPa and 34.88 kPa, respectively. This result indicates that the loads that the connections with the meshes could withstand were larger than those in other cases. Therefore, the mechanical performance of the flexible material is more competitive than that achieved by traditional roughening with aggregates.

Another significant difference could be found in the shape of the plots presented in Figure 6. For Cases N and A, the level of strain within quasi-elastic behavior was very small, less than 0.002. After this, the system failed, and it was unable to bear any relevant load. The performance is clearly fragile with sudden failure. For the straight mesh (A-SM), quasi-elastic behavior occurred under very small strains of only 0.004, the same order as in the other cases; however, failure did not occur. A slow hardening took place until a peak average stress of 74.07 kPa was reached, and then a softening of the system took place until failure with strain close to 0.046 (11 times) occurred. The failure was progressive, not sudden; it involved a mechanism that allowed the dissipation of more energy (larger area under the curve). The inclined mesh (A-IM case) displayed similar behavior in the beginning, including nonlinear hardening until a peak stress of 61.94 kPa was reached, with a strain level of 0.01; from this point, the system was capable of maintaining the same level of stress for an impressive failure strain of 0.11 (11 times). This indicates pseudo-plastic behavior that dissipates a great deal of energy compared to the other systems, because the area under the plot is much larger. It is clear that there is a tightening effect of the mesh over the concrete that produces a friction mechanism.

For both samples with mesh connections, the material had a positive effect. During the loading process, after the peak load, several complex types of damage took place: cracking of the concrete, debonding of the tows and the concrete, mechanical locking of the concrete in the mesh gaps, and the breakage of threads. Nevertheless, for the A-SM, damage continued to grow, and the system was unable to withstand the applied load, losing stiffness and producing a softening effect. On the contrary, the A-IM was capable of maintaining the same load level. The reason for this pseudo-plasticity lies in the mesh position.

The inclined mesh has several advantages compared to the straight mesh. Firstly, the contribution of fibers to the strength comes from a tightening effect. The strength of fibers is very high compared to the levels of load, so its contribution to the load-bearing is negligible. Hence, the placement in a non-optimal direction (in terms of only the stress response) is not a limitation. Secondly, straight fibers are distributed discontinuously; there are strands only every 25 mm and, hence, there are important gaps with no fibers. Moreover, the flexibility of the perpendicular fibers cannot produce an effective locking mechanism in the gaps. On the contrary, the inclined mesh covers the whole width with its strands and does not leave any empty spaces without fibers. Moreover, the inclined threads can produce a more effective locking mechanism in the gaps. Hence, the inclined mesh provides a uniform response that helps to maintain a high level of friction.

Compared to push-out tests by other authors like Zou et al. [26], Gong et al. [43], Di et al. [18], or Rajchel et al. [25], the achieved ultimate loads of connection present less strength capability. The reason for this relies on two things. Firstly, the flange of a profile is much stiffer and has greater strength than the thin CFRP used in our samples. Therefore, push-out tests with bolt connections will always bear much more load. Secondly, bolts are made of steel and can experience large strains and stresses before failure compared to a GFRP mesh. Therefore, from the strength point of view, the systems are not comparable. Moreover, we note that Gong et al. [43] found a stress level of 0.27 MPa for natural bond between an SIP system and concrete, which is 10 times the level of our stresses; this could be justified by the increase in friction produced by the bolt prestress. On the other side, the proposed system in this study does not use any steel, reducing the risk of future corrosion, and it is very simple to manufacture, achieving thinner slab solutions. These advantages can make it competitive in the market.

## 4. Finite Element Method Simulations

In order to better understand the performance of the mesh connections, some finite element simulations were performed in order to clarify the locking mechanism and the shear response of the interface between the concrete and the CFRP sheet. Both the A-SM and A-IM cases were modeled using a general-purpose finite element software program, ABAQUS (2017, Dassault Systems, Pawtucket, Rhode Island, U.S.) [46]. In the following paragraphs, assumptions about the geometry, materials, contacts, mesh, and boundary conditions are presented.

### 4.1. Geometry

The general geometry of all numerical models was the same, and it corresponded to the shortest concrete block. The concrete block and the thin CFRP sheet were considered independent solids connected at the interface surface. Glass fiber materials were modeled as beam elements with the same sections as the primary material. These were added in the corresponding direction (orthogonal or inclined at 45°). Details of the aggregates and the thin layer of resin or glass fiber mesh–CFRP sheet strip connector were not explicitly represented in the model and were implicitly considered in the contact between surfaces.

### 4.2. Materials

The CFRP sheet was under a low level of stress compared to the ultimate capacity of the material. Therefore, the material composition was defined as linearly elastic. Generally speaking, the block of concrete was not damaged; only the contact zone with the different connection systems had cracks. As this zone was limited to the contact surface, it was treated separately. Hence, the concrete block was also defined as a homogeneous, linearly elastic material.

Glass fiber fabrics are defined as elastic–plastic materials. The definition of the elastic branch is based on the provider’s properties, including a Young’s modulus of 72 GPa and a Poisson coefficient of 0.3. The plastic properties of the glass fiber materials were used to fit the models, and these were intended to include all the complex effects of the three-phase connection between the concrete, fabric, and CFRP sheet: sliding of the fibers, debonding of the fabric–CFRP sheet connection, and cracking of concrete around the fabric, for the cases including glass fiber meshes. The included parameters were the yielding stress of 180 MPa and consistent mass matrix formulation to obtain the shear modulus for the case of the straight mesh (A-SM), and a yielding stress of 25 MPa, transversal stiffness of 650 MPa, and slenderness compensation value of 0.9 for the definition of the plastic behavior of the A-IM cases. All these parameters were considered carefully as they are essential for the correct fitting of the experimental results presented later on.

The Young’s modulus of the CFRP was experimentally obtained by testing the samples, while the Poisson’s coefficient was taken from Munoz et al. [47]. The Young’s modulus of the concrete was calculated from an empirical equation of Eurocode 2 [48] that relates the compressive strength $f_{ck}$ to the Young’s modulus; see Equation (1).

The average concrete compressive strength was taken from Table 1, which gives a value of 20.4 MPa. The Poisson’s coefficient of concrete was derived from the general properties of Eurocode 2 [48]. Table 2 summarizes the mechanical properties of the CFRP and concrete.
(1)$$E\sim 22,000{\left(\frac{f_{ck}+8}{10}\right)}^{0.3}$$

### 4.3. Contacts

The surface contact between CFRP and concrete is a complex system. Concrete cracks, and the block also generates friction with bonded aggregates. Moreover, the mesh introduces a complex tightening system that modifies the performance dramatically, as was observed in the experiment. Therefore, modeling the details of the interaction of these elements is beyond the scope of this work; additionally, there is no experimental evidence to sustain any assumption. Hence, to model this complex interface, we used a strategy of defining contacts. This approach simplifies the model but does not lose generality, and it is supported by the empirical observations. Hence, aggregates were not considered physically in the simulation, but they indirectly participate when adjusting contact properties to the experimental observations.

Two models were used to represent the concrete–CFRP sheet interface; the first was a cohesive zone model (see Elices et al. [49]) including damage parameters that required the definition of the stiffness coefficient, maximum nominal stress, total displacement, and an exponential parameter. The values are not included here because it was not possible to reach a proper adjustment between the model and the experimental curve. Therefore, this model was considered inappropriate for our experiments. The second model was a frictional model defined by its tangential behavior based on the friction coefficient; see Schellart [50]. For both simulated experiments A-SM and A-IM, the friction coefficient was taken as 0.1. Undoubtably, this value must be taken with caution. However, in the discussion of the numerical results, it is possible to observe the consistency of the model.

The connection of the glass fiber mesh was defined as an embedded restraint in the concrete and CFRP sheet solid elements.

### 4.4. Mesh

The concrete was modeled by linear hexahedral elements of type C3D8R—an 8-node, brick-shaped, three-dimensional element with one point of integration to reduce the computational cost. According to the assumptions, the modeling of the concrete did not require precision. The CFRP sheet was modeled by linear wedge elements of type C3D6. This element type has 6 nodes and 2 integration points. The glass fiber mesh was modeled with B31 truss elements. The concrete, CFRP, and glass fiber mesh were meshed dependently with a maximum seed dimension of 15 mm. The Hex-Sweep technique was used. If a region is swept meshable, Abaqus/CAE can generate the swept mesh on a region that has been assigned the Hex.

The resulting mesh can be observed in Figure 7.

### 4.5. Boundary Conditions

The boundary conditions were the same for both models. The bottom surface of the CFRP was simply supported. The shear load was indirectly imposed by a uniform displacement along the specimen’s longitudinal direction for all experimentally loaded faces. Finally, the CFRP face that was coplanar with the concrete face where the displacement was imposed was fixed.

The analyses were carried out using a static general procedure, obtaining the shear load as the reaction associated with the imposed displacement for all time increments. The total time-period was assumed to be one second and the maximum number of increments was defined as 100.

Figure 8 shows the boundary conditions and imposed displacement used in the models.

## 5. Numerical Results and Discussion

### Numerical Results

For the assumed approach of combining a frictional model to represent the CFR–concrete interaction together with embedding the mesh to model the CFRP–mesh–concrete response, there is a consistent relationship with the achieved results. The inclination of the mesh (A-IM) allowed us to consider the tensile response of the wires of the glass fiber fabric because of their relative orientation with respect to the load. A part of this load is axially resisted by the wires, which progressively develop a nonlinear response. This process allowed the load to continuously increase, although at progressively reduced stiffness up to failure. In contrast, the mesh only contributes to resisting the applied load through the shear mechanism for the A-SM cases. Hence, when the maximum shear strength is reached, all mesh capacity is lost, and only the frictional connection between the CFRP and concrete is available to resist the applied shear load. This theoretical analysis is translated in the experimental and numerical simulation results into the clear maximum peak of the force, followed by a force decrease, culminating in a constant residual shear strength due to the concrete–CFRP sheet frictional interaction.

It should be noted that the obtained shear stiffness of the specimen is very similar in both cases (A-SM and A-IM), as is the slope after yielding. However, the yielding point is, in fact, influenced by the resisting mechanism of the embedded mesh, being lower for the combined axial-shear case (A-IM) than for the pure shear case (A-SM).

The numerical results in terms of force reaction vs. displacement are compared to the average curve of the experimental results for each case. It was observed that the proposed numerical model accurately predicts the mechanical response of the experimental tests for both mesh orientations.

No numerical values of the fitting quality are provided because the ad hoc definition of some of the variables of the numerical model would invalidate this quantification. The fitting quality can be observed through the comparison of the full force–displacement curves in Figure 9.

The theoretical behavior inferred for both novel connection systems considering the embedding of a flexible glass fiber mesh can be seen in Figure 10. It is a trilinear curve with an elastic branch and a non-elastic short hardening up to the peak load. No significant difference in the qualitative response of these two first branches was noticed between the two considered cases. Nevertheless, after the peak load, the inclined mesh produces a hardening branch with additional energy dissipation until failure. The straight orientation of the mesh produces a softening branch after the peak load, followed by a residual constant strength.

Although further research is necessary to accurately fit the defining properties of these models (calculation of the main points), the description of the qualitative response in Figure 10 is the first necessary step to properly focus future research on this novel connection system for thin FRP–concrete hybrid structures based on the embedding of a continuous fiber fabric.

## 6. Conclusions

In this research, an experimental study and numerical analysis of CFRP–concrete hybrid elements subjected to in-plane interface shear tests were carried out. The mechanical responses of different CFRP–concrete connection strategies, including a novel proposal based on embedding a fiber material, were analyzed and discussed. A simplified numerical model was fitted with the experimental results. The following concluding remarks can be made:Specimens without connectors (N) and those with only aggregate connections (A) presented the weakest shear resistance. The performance was fragile in all cases, with sudden failure. The post-critical response was related to merely friction between materials, not allowing significant energy dissipation. Both connections are considered unreliable for real works, disregarding even the classical connection approach of increasing the roughness via aggregate adhesion.Embedding a flexible structural glass fiber material as a connector between the CFRP sheet and the concrete increased the shear load-bearing capacity and enhanced the post-peak shear strength, changing the failure mode from sudden to progressive. Moreover, the system with the mesh connection could dissipate much more energy than that with the aggregate connection. The curve performance of the inclined mesh showed some similarities to a bolt connection.The numerical simulations combining a frictional model and the embedding of a fabric were capable of reproducing the experimental curves. Adjustment values must be taken with caution but are consistent with the results.The numerical model explains that the orientation of the fabric plays an important role in the structural response because of the possibility of combining two resisting strategies, axial and shear, when inclined with respect to the load direction.A real locking mechanism is a complex combination including concrete cracking, debonding of fibers and concrete, sliding of fibers, locking concrete in the grid gaps, friction between concrete and FRP, and breakage of fibers. The proposed strategy simplifies this complex combination of resisting mechanisms into only two equivalent interfaces: the frictional interface between concrete and FRP and the embedded beam elements between concrete, mesh, and FRP.The achieved level of load is less than that for a traditional bolt system but it has the advantage of not using steel connectors, and it involves a very simple manufacturing process for thinner slabs.

## Figures and Tables

**Figure 1 materials-14-05184-f001:**
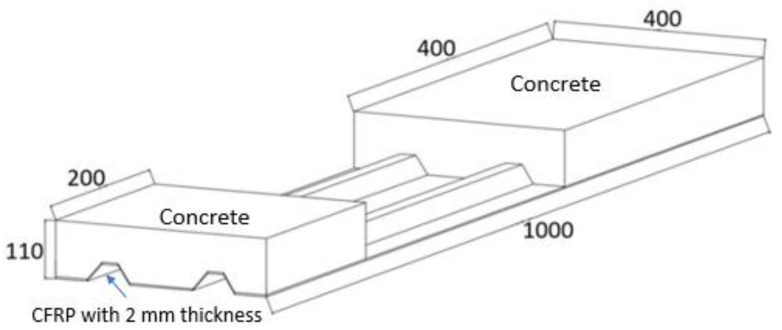
Geometry of the specimens (dimensions are in millimeters).

**Figure 2 materials-14-05184-f002:**
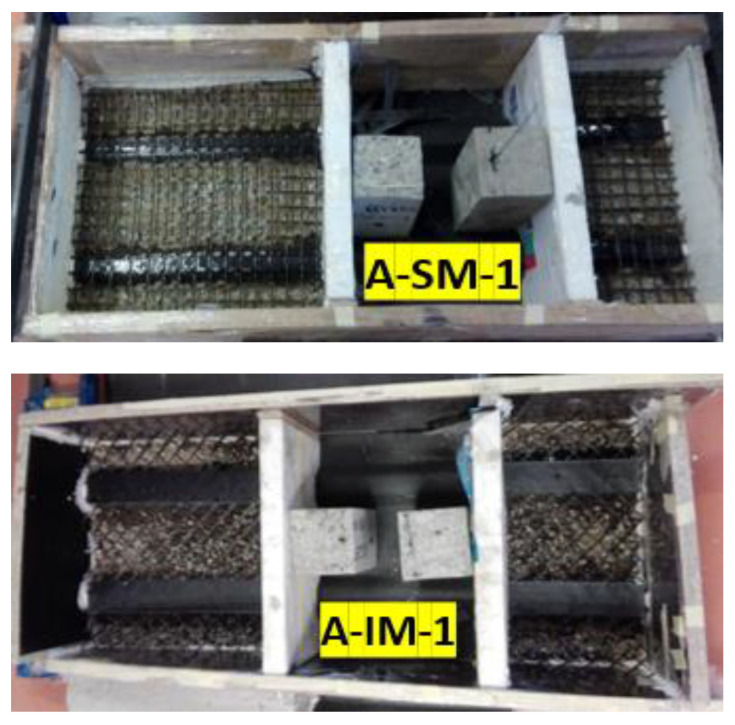
Formwork and CFRP connection of specimens A-SM-1 and A-IM-1 before concrete was poured.

**Figure 3 materials-14-05184-f003:**
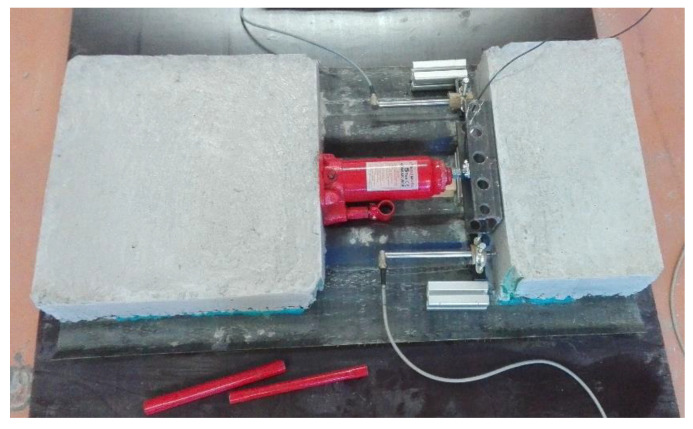
Shear test setup.

**Figure 4 materials-14-05184-f004:**
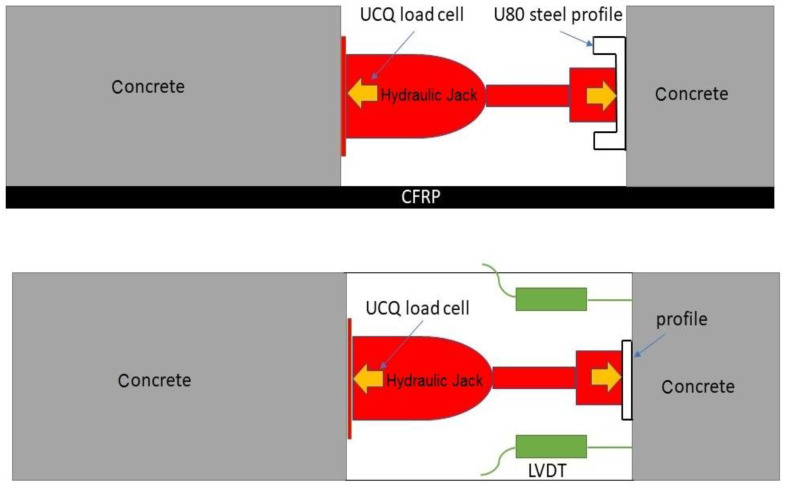
Test setup and instrumentation. Side view and top view.

**Figure 5 materials-14-05184-f005:**
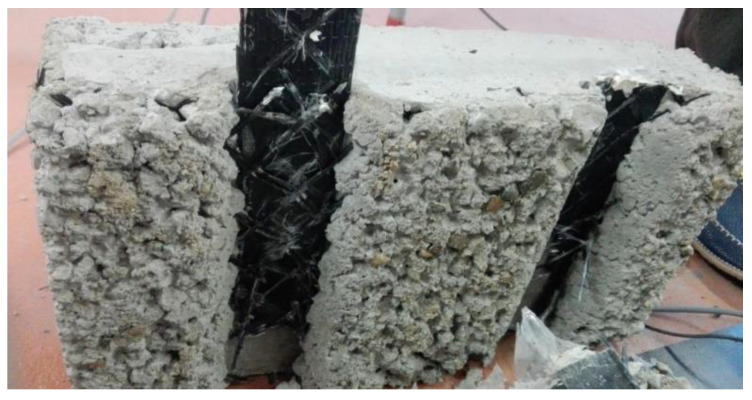
Failure mode of A-IM cases combining mesh debonding from the CFRP sheet and mesh breakage.

**Figure 6 materials-14-05184-f006:**
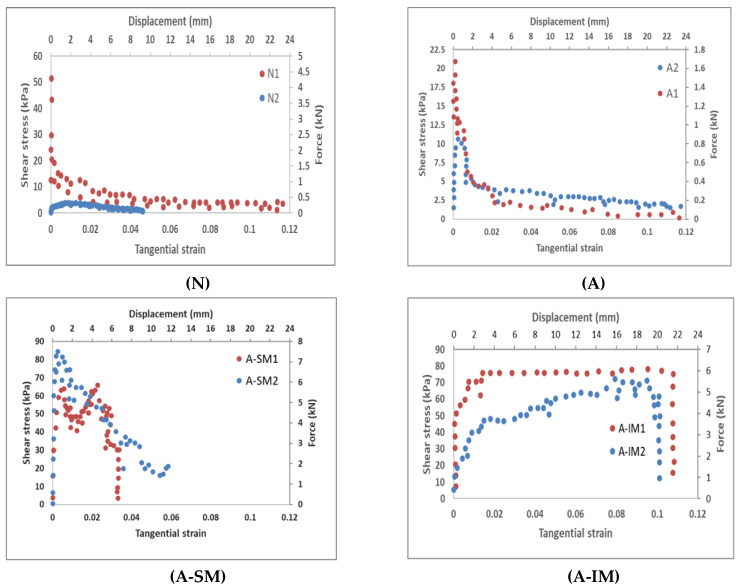
Experimental results: force vs. relative displacement of concrete with respect to CFRP and shear stress vs. average tangential strain of the connection area.

**Figure 7 materials-14-05184-f007:**
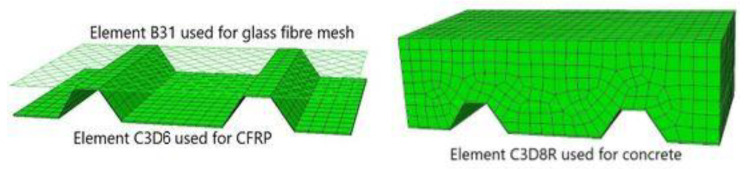
Mesh of the CFRP sheet and glass fiber fabric (**left**) and mesh for the concrete block (**right**), Case A-IM.

**Figure 8 materials-14-05184-f008:**
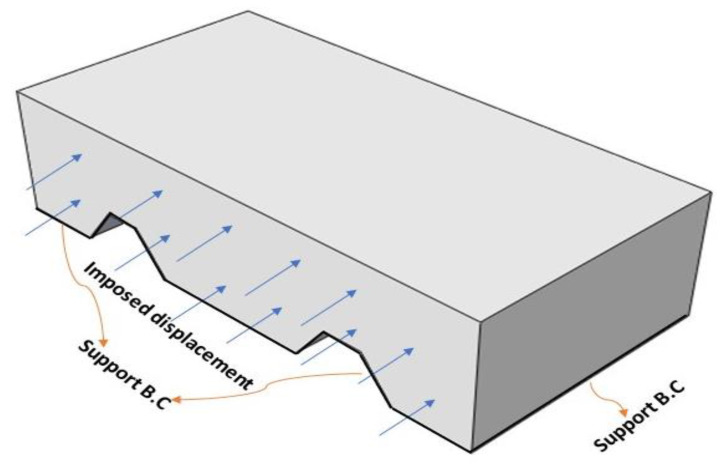
Geometry and boundary conditions.

**Figure 9 materials-14-05184-f009:**
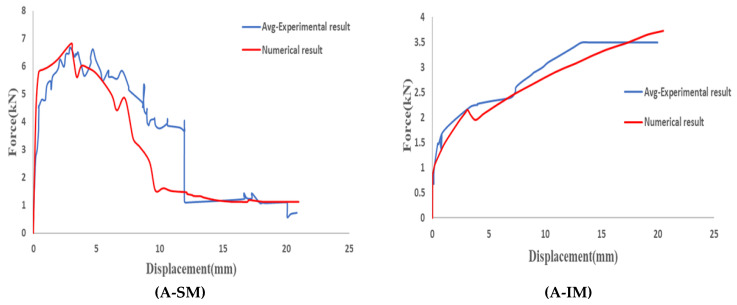
Force–displacement plots for the numerical models and experimental specimens.

**Figure 10 materials-14-05184-f010:**
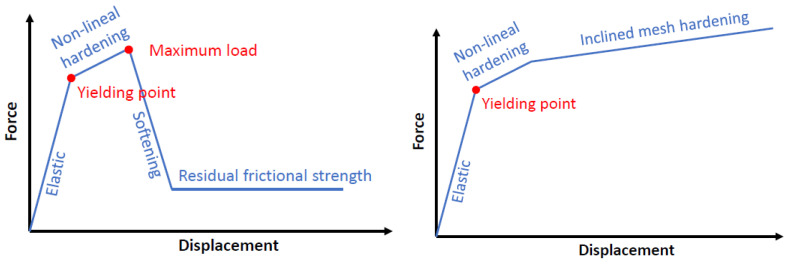
Idealization of the behavior of a straight (**left**) mesh connection and an inclined (**right**) mesh connection.

**Table 1 materials-14-05184-t001:** Specimen labelling and concrete strength.

Name	Aggregates	Straight Mesh (0°)	Inclined Mesh (45°)	Compressive Strength (MPa)
N-1	-	-	-	22.0
N-2	-	-	-	22.6
A-1	Y	-	-	18.9
A-2	Y	-	-	19.0
A-SM-1	Y	Y	-	19.0
A-SM-2	Y	Y	-	20.8
A-IM-1	Y	-	Y	20.0
A-IM-2	Y	-	Y	21.0

Y: Yes.

**Table 2 materials-14-05184-t002:** Mechanical properties of the concrete, CFRP, and glass fabric for numerical simulation.

Material	Young’s Modulus (MPa)	Poisson’s Coefficient
Concrete	29,975	0.20
CFRP	45,500	0.26
Glass Fabric	72,000	0.30

## Data Availability

MDPI Research Data Policies.

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
