# Peer review of "Experimental and Numerical Study of Shear Interface Response of Hybrid Thin CFRP–Concrete Slabs"

_materials, 2021, doi:10.3390/ma14185184_

Round 1

Reviewer 1 Report

Dear Authors,

please see the attached file.

Reviewer 2 Report

The manuscript entitled: “Experimental and numerical study of shear interface response of hybrid thin CFRP–concrete slabs” is relevant for the Materials journal. It based on original research supported by computer modelling. The main question addressed by the research is connected with composites.

The article is relevant and interesting, but the subject area is not sufficiently compared with other published material. The manuscript requires to be supplemented at discussion part. The provided research should be compare with results given in other up-to-date literature. Moreover, some practical application for new composite should be described. The lack of discussion is a serious flaw of this manuscript.

Overall the paper is well written but it requires some minor changes such as:

  • Authors: please add “*” behind the corresponding author.
  • Introduction: line 24, please start line on beginning (without space).
  • Introduction: line 40 – please verify references (wrong numbers).
  • Introduction: line 46 – please verify references (wrong numbers).
  • Introduction: lack of references for literature in points 9 and 10.
  • Introduction: line 66 – please verify references (wrong citation style – 2 authors).
  • Introduction: line 80 – please verify references (wrong citation style – 2 authors).
  • Experimental results…: line 227 – wrong number of figure; there is no plot on this picture.
  • 4.2. Materials: please add information if aggregates were taken into consideration during modelling and add proper information.
  • References: please use a style coherent with journal template.

The conclusions part consistent with the evidence and arguments presented in the manuscript part and address the main question posed.

Reviewer 3 Report

The following comments and suggestions will improve the understanding of the material presented by this paper.

Review comments

  • The authors should explain why the used such a small size of aggregates and if this plays a role in the interface.
  • How the ultimate tensile strength of the CFRP sheets was measured?
  • There are two Figures 1. Please correct it.
  • In Figure 6, all the subplots should have the same limits in the x-coordinate. Also a line connecting the dots for each experiment would be useful.
  • A picture showing the formwork and CFRP connection for A-IM-1 should be added (like the one with A-SM-1).
  • The empirical equation used in Eurocode 2 (line 294) should be presented.

Round 2

Reviewer 2 Report

The manuscript was significantly improved, however it requires still some minor revisions and text veryfication. It is not fully coherent with template, there is lack of information for example about COI or authors contributions. The referencees need to be editing, the format is slightly different than given in template. In the text are some gaps, for example in line 405 - lack of proper reference ("Error! Reference source not found. ").
